# Association between Maternal Education and School-Age Children Weight Status: A Study from the China Health Nutrition Survey, 2011

**DOI:** 10.3390/ijerph16142543

**Published:** 2019-07-16

**Authors:** Yuejing Feng, Lulu Ding, Xue Tang, Yi Wang, Chengchao Zhou

**Affiliations:** 1School of Public Health, Shandong University, Jinan 250012, China; 2NHC Key Laboratory of Health Economics and Policy Research, Shandong University, Jinan 250012, China

**Keywords:** maternal education, weight status, BMI, children

## Abstract

Childhood abnormal weight status is a global public health issue. This study aims to explore the association between maternal education and weight status of school-age children using the data from the China Health Nutrition Survey (CHNS), wave 2011. Body Mass Index (BMI) is calculated based on children’s weight and height and is divided into three levels (normal, underweight, overweight/obesity). Logistic regression is used to assess the relationship of maternal education and children weight status. The prevalence of childhood underweight and overweight/obesity are 5.9% and 21.7%, respectively. Children with high maternal education are found less likely to be underweight (technical college: Odds Ratio (OR) = 0.223, 95% Confidence Interval (CI) = 0.052–0.956, above college: OR = 0.182, 95% CI = 0.041–0.812). Children with maternal education at junior high school are of 2.170 greater odds to be overweight/obese than those with maternal education at junior high school and below (OR = 2.170, 95% CI = 1.398–3.370), and children with maternal education at technical college are of 2.397 greater odds to be overweight/obese than those with lower education level (OR = 2.397, 95% CI = 1.478–3.887), and children with maternal education at above college are of 2.146 greater odds to be overweight/obese than those with lower education level (OR = 2.146, 95% CI = 1.293–3.560). A significant association between maternal education and children’s weight status is found. Targeted interventions for mothers with different education levels should be carried out to effectively manage the children’s weight status.

## 1. Introduction

Overweight, obesity, and underweight are three forms of weight imbalance that seriously endanger human health. In 2018, about 1.9 billion people worldwide were overweight and 462 million were underweight. Being overweight/obese and underweight are conditions that simultaneously exist in children, which have become a double burden that threatens the well-being of children in developing countries [1,2]. About 45 percent of deaths among children under five years old are related to malnutrition, mainly occurring in low-and middle-income countries [3]. The prevalence of overweight/obesity among children and adolescents has also risen dramatically from four percent in 1975 to over 18 percent in 2016 [4].

Unusual weight is a key risk factor for health status, especially for children and adolescents. Various acute or chronic diseases in the adult stage are long-term consequences of weight imbalance in childhood [5], which makes it extremely urgent to take measures to cope with abnormal weight status in children. Several previous studies indicate that being underweight is associated with children’s health outcomes [6,7]. Childhood overweight and obesity can lead to many complications [8]. It threatens the cardiovascular system, endocrine system, respiratory system and liver, skeletal development, psychological behavior and cognitive intelligence. Hypertension, diabetes, asthma and psychological problems are typical hazards to children’s health [9].

Parental characteristics are one kind of the most important determinants of children’s health behaviors and development [10]. Some previous studies found that maternal characteristics have a stronger effect on children’s health status than paternal characteristics do [11,12,13]. Mothers play a major role in children’s health outcomes in the growth of children. As an important characteristic, maternal education level has been demonstrated to be a vital determinant of children’s health status [11,14,15,16]. Several studies interpreted that mothers with lower education levels feed children with more unhealthy foods [13,17], which may negatively influence children’s health status. A study finds that mothers with higher education are more likely to initiate early breastfeeding for the new born babies [18]. Exploring the relationship between mothers’ education level and children’s weight might be beneficial to the health management of children’s weight.

Previous studies have demonstrated that children’s weight status was associated with parental characteristics, but mainly focusing on the combined effect of both fathers and mothers [19,20]. In addition, a large number of studies in China only explored unusual weight status and its determinants among preschool aged children [21,22,23,24,25,26]. To date, however, few studies have examined the association between weight status and maternal education among school-age children. This study aims to examine the association between school-age children’s weight status and maternal education.

## 2. Materials and Methods

### 2.1. Data

The data used in this study are from the China Health Nutrition Survey (CHNS). CHNS is an ongoing follow-up survey project that conducted by the China Center for Disease Control and Prevention and the University of North Carolina. The purpose of the CHNS survey is to explore the impact of China’s economic transformation and family planning policies on resident’s health and nutritional status. The content covers demographic characteristics, economic development, public resources and health indicators and information on food markets, medical institutions, etc. The first round of the CHNS was conducted in 1989, and eight follow-up panels were conducted in 1991, 1993, 1997, 2000, 2004, 2006, 2011, and 2015. The wave of 2011 data was employed in the current study. Interested readers are encouraged to see the website for details [27].

### 2.2. Study Design and Sample

A multistage, random cluster sampling method was employed in CHNS. Counties in the sample areas were stratified by income (low, middle, and high). Villages and townships within the counties and urban and suburban neighborhoods within the cities were selected randomly. A total of approximately 7200 households with over 30,000 individuals in 15 provinces and municipal cities were eventually included (See Appendix A
Figure A1). The survey is mainly divided into two parts: questionnaire survey and physical measurement. The data from the questionnaire was collected by respondent’s self-reported answers. Since the 2015 wave, data do not involve some variables we intend to analyze in this study, and the latest available data of wave 2011 were filtered with children aged 7–18 years old to match with corresponding mother’s information manually based on the individual ID and participants’ relationship. Those children who missed key information (e.g., weight or height, maternal education) were removed. Finally, a total of 1081 paired mother-children participants were included in analysis.

### 2.3. Variables

Body Mass Index (BMI, kg/m^2^) is calculated based on weight and height followed calculation formula standard. Underweight and overweight/obesity cut-off values of different ages refers to the classification criteria of body weight index for 5–19 years old boys and girls (z-scores). It is considered as underweight if BMI < Mean −2 Standard Deviation (SD), overweight if BMI > Mean + 1SD and obesity if BMI > Mean + 2SD. It is defined as 0 if BMI is normal, underweight as 1, overweight or obesity as 2.

Education of mothers is divided into four levels: junior high school and below, high school, technical college, and above college. In addition, children’s weight status, maternal education, children gender, region, maternal employment status, household incomes are also included into analysis in this study.

### 2.4. Statistical Analysis

The Chi-square test is used to compare the prevalence of underweight, overweight/obesity across different groups. Two binary multivariate logistic regression models are employed to examine the association between maternal education and children’s weight status. We set children’s BMI classification (underweight, normal, overweight/obesity) as independent variable (Y) and set dummy variables for underweight and overweight/obesity. In model 1, only maternal education level was included in the regression. In model 2, we included additional factors of children gender, maternal age, maternal employment status, and household income into model as covariates to further assess the association between maternal education and children weight status. All data analyses were employed using SPSS 24.0 (IBM Corporation, Armonk, NY, USA) and the significance level was set at 0.05.

## 3. Results

### 3.1. Basic Characteristics of the Children

A total of 1081 children aged 7 to 18 years old were included in the analysis. Of whom, 48.4 percent are girls and 61.1 percent of the children are from rural area. The mean weight and height are 41.40 (SD = 15.01) kg and 147.13 (SD = 17.11) cm, respectively. Table 1 shows the prevalence of normal weight, overweight and obesity in detail. Table 2 shows the basic information of the mothers. The majority of the maternal education level is junior high school and below, accounting for 65.68 percent. About 78.35 percent of the mothers are employed.

### 3.2. Children’s Weight Status Distribution across Different Groups

Table 3 presents the results of univariate analyses. The prevalence of underweight (3.5%) and overweight/obesity (13.5%) among the boys are both higher than those in girls (2.4%, 8.3%). The prevalence of underweight is higher (6.68%) in the children with age of 11 to 14 years, and overweight/obesity rate is higher (25.76%) in the children of 7 to 10 years. A higher prevalence of overweight/obesity is found among children with maternal education of technical college (29.57%) and above college (31.87%), comparing with children with maternal education of junior high school and below (16.14%) and high school (17.32%) (see Figure 1). In addition, children from lower income (Q1) households have a higher prevalence of being underweight than those from higher income households, while the results of the overweight/obesity contingency are the opposite to those of the underweight.

### 3.3. Effect of Maternal Education on Children’s Weight Status

We use two models to examine the association of maternal education and children’s weight status (see Table 4). In model 1, we only include maternal education in the analysis. The result shows that compared with those whose mothers have an education level of junior high school and below, the children with a maternal education level of high school are of 2.065 greater odds to be overweight/obesity (OR = 2.065, 95% CI: 1.370–3.113), and those children with technical college are of 2.357 greater odds to be overweight/obesity (OR = 2.357, 95% CI: 1.506–3.690), while above college are of 2.251 times (OR = 2.251, 95% CI: 1.463–3.464). In contrast, the higher the maternal education is, the odds that the children will be underweight will be lower (OR < 1). In model 2, we included some other potential confounding factors (children gender, maternal age, etc.), the association between maternal education and children’s weight status is still statistically significant. In addition, factors including gender, age, and maternal employment are associated with the children’s weight status.

## 4. Discussion

According to the Report on Nutrition and Chronic Diseases of Chinese Residents, the overweight and obesity rates of children and adolescents aged 6 to 17 in China in 2012 were 9.6%, 6.4%, and the average rate of underweight was 9.0% [28]. The current study shows that the prevalence of underweight is 5.9 percent, which is lower than the national level in 2012. The prevalence of overweight/obesity is 21.7 percent, which is significantly higher than the national level in 2012 and is also higher than the estimated prevalence in some previous Chinese studies in China [29,30,31,32]. The overweight/obesity rates among the boys and girls are 13.41 percent and 8.65 percent, respectively, and the prevalence among the boys is significantly higher than that in the girls, which is consistent with previous studies [33,34,35]. This increasing trend of overweight/obesity among school-age children is serious, and is a priority we should focus on.

The present study finds that maternal education is associated with children’s weight status. The school-age children whose mothers have a higher education level would be more likely to be overweight/obesity (*p* < 0.01). A study by Liu also finds that higher maternal education is associated with higher rates of children’s obesity [36]. Many studies in western countries have pointed out that the education level of mothers is negatively correlated with the prevalence of overweight and obesity among children [37,38]. Interestingly, the association between maternal education and children’s weight status in China is different from that in western countries and the maternal education and children weight outcomes of a high-income country cannot directly be transferred to a middle-income country.

Some previous studies indicate that a higher level of maternal education is associated with a better dietary quality and a more adequate intake of various nutrients for children [39]. Those children coming from well-educated families consumed more animal protein and fat and are more likely to have higher serum levels of the lipoprotein compared with their counterparts [40]. In addition, the children with higher maternal education level are found to have lower physical activity and longer time for sedentary lifestyle [39,41,42,43,44]. In addition, the difference in the overweight and obesity pattern between China and Western countries might be attributed to China’s unique culture and lifestyle. Different social and cultural environments can influence mothers’ perceptions of children’s body shape [45]. Children are the inheritance of the next generation in a Chinese family and are often beloved. Parents, especially mothers would give their children as much as they can. Thus, overweight/obesity may be an outcome of overnutrition. The China Obesity Report in 2017 also stated that in developed countries, children with low socioeconomic status (SES) have higher rates of obesity than children with higher SES. In China, the obesity rate among children with high parental—or at least maternal—education is higher than that of lower parental education, and the sociocultural factors remarkably influence the development of obesity [46].

In this study, boys are found more likely than girls to be overweight and obese after controlling for potential covariates, which is consistent with many previous studies [22,31,35]. The girls have a strong sense of weight and body shape, and then have better self-awareness for body shape management than the boys [47]. It is necessary to educate children to correctly understand their own body shape from the perspective of as caregivers who are mainly mothers.

It is found that the children whose mothers are employed are less likely to be overweight/obese. Similar to the study by Xie et al., our study also finds that maternal employment has a significantly negative effect on the likelihood of overweight status/obesity among school-age children [48]. Similar to our finding, this previous study showed that maternal employment status has a significant impact on the nutritional status of children under 5 years old in rural areas and children whose mothers work outside have the worst nutritional status compared with local farmer mothers [49]. Employed mothers have less time to pay attention to their children’s reasonable nutritional intake and healthy weight management than unemployed mothers. The employed mothers spent less time on meal preparation and were more reliant on substitute childcare due to time constraints, as has been documented by Oddo et al. [50].

Obviously, the present finding should be an impetus to take targeted measures in mothers with different education levels. It is urgent to strengthen the health education of mothers so that they can correctly understand the importance of proper nutrition intake and weight management of children. High attention about overweight and obesity, especially for those mothers with high education level, should be prioritized to reduce the potential damage due to the imbalanced weight of their children. Future research should address the impact of maternal socioeconomic status on children’s weight status in order to discover more maternal characteristics associated with children’s health. Future research should address the impact of maternal socioeconomic status on children’s weight status in order to discover more maternal characteristics associated with children’s health. For the purpose of improving the maternal cognitive level regarding children’s reasonable nutrient intake and body shape, future programs that focus on well-targeted health education for mothers should be implemented. It is also necessary to evaluate intervention effects to further strengthen child weight management.

This study is based on a nationwide survey and is representative of maternal education and children in the whole of China. There are some limitations in this study. First, we only focused on school-age children in this study, and the results may not be generalized to adults and pre-school age children. Second, our study only included limited variables when exploring the relationship of maternal education level on children’s body weight, which may be remedied in follow-up studies. Third, our study had a cross-sectional design, and the results could not be interpreted as cause and effect. Finally, the data were mainly based on self-reported measures, which might lead to possible recall biases.

## 5. Conclusions

The prevalence of overweight/obesity among school-age children is rather high. This study demonstrated a significant association between maternal education and children’s weight status. School-age children with a higher education level are more likely to be overweight/obese but less likely to be underweight. In addition, factors including gender, age and maternal employment status are also associated with children’s weight status. Early intervention is required to manage the weight imbalance among school-age children.

## Figures and Tables

**Figure 1 ijerph-16-02543-f001:**
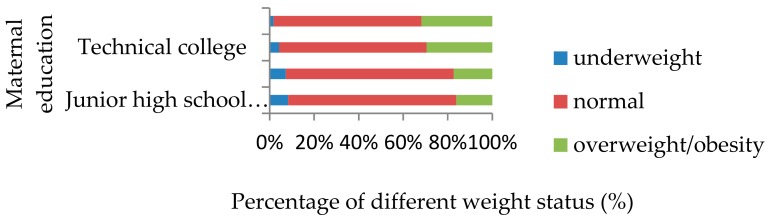
Children weight status distribution of different maternal education level.

**Table 1 ijerph-16-02543-t001:** Basic characteristic of the school-age children.

Characteristics	Observations or Mean	%
Residence		
Urban	421	38.9
Rural	660	61.1
Gender		
Boys	558	51.6
Girls	523	48.4
Age (years)		
7–10	396	36.6
11–14	419	38.8
15–18	266	24.6
Weight (kg)	41.40	-
Height (cm)	147.13	-
Weight Status		
Normal	782	72.3
Underweight	64	5.9
Overweight	145	13.4
Obesity	90	8.3
Total n (%)	1081	100

**Table 2 ijerph-16-02543-t002:** Basic characteristics of the mothers.

Characteristics	Observations	%
**Education**		
Junior high school and below	710	65.7
High school	14	13.1
Technical college	108	10.0
College and above	121	11.2
Employment		
Yes	847	78.4
No	234	21.7
Occupation		
No occupation	234	21.7
Professional and technical worker	111	10.3
Manager/office staff	93	8.6
Farmer	225	20.8
Worker	143	13.2
Service/driver	215	19.9
Others	60	5.6

**Table 3 ijerph-16-02543-t003:** Children’s weight status across different subgroups.

Variables	*N* (%)	Weight Status (n, %)			*X* ^2^	*p*
		Underweight	Normal	Overweight/Obesity		
**Residence**					2.081	0.353
Urban	421 (39.0)	21 (5.0)	301 (71.5)	99 (23.5)		
Rural	660 (61.1)	43 (6.5)	481 (72.9)	136 (20.6)		
Gender					15.315	<0.01
Boys	558 (51.6)	38 (6.8)	375 (67.2)	145 (26.0)		
Girls	523 (48.4)	26 (4.97)	407 (77.8)	90 (17.2)		
Children age (year)						
7–10	396 (36.6)	24 (6.1)	270 (68.2)	102 (25.8)	21.462	<0.01
11–14	419 (38.8)	28 (6.7)	291 (69.5)	100 (23.9)		
15–18	266 (24.6)	12 (4.5)	221 (83.1)	33 (12.4)		
Maternal age (year)					11.369	0.023
27–36	358 (33.1)	22 (6.2)	239 (66.8)	97 (27.1)		
37–46	664 (61.4)	41 (6.2)	497 (74.9)	126 (19.0)		
47–57	59 (5.5)	1 (1.7)	46 (78.0)	12 (20.3)		
Maternal education					36.202	<0.01
Junior high school and below	710 (65.7)	54 (7.6)	536 (75.5)	120 (16.9)		
High school	142 (13.1)	6 (4.2)	94 (66.2)	42 (29.6)		
Technical college	108 (10.0)	2 (1.9)	71 (65.7)	35 (32.4)		
College and above	121 (11.2)	2 (1.65)	81(67.0)	38 (31.4)		
Maternal employment					5.657	0.059
Yes	847 (78.4)	50 (5.9)	626 (73.9)	171 (20.2)		
No	234 (21.7)	14 (6.0)	156 (66.7)	64 (27.4)		
Maternal occupation					31.524	<0.01
No occupation	234 (21.7)	14 (6.0)	156 (66.7)	64 (27.4)		
Professional and technical worker	111 (10.3)	3 (2.7)	73 (65.8)	35 (31.5)		
Manager/office staff	93 (8.6)	5 (5.4)	58 (62.4)	30 (32.3)		
Farmer	225 (20.8)	18 (8.0)	174 (77.3)	33 (14.7)		
Worker	143 (13.2)	6 (4.2)	112 (78.3)	25 (17.5)		
Service/driver	215 (19.9)	14 (6.5)	165 (76.7)	36 (16.7)		
Others	60 (5.6)	4 (6.7)	44 (73.3)	12 (20.0)		
Household income ^a^					13.448	0.036
Q1	271 (25.1)	20 (7.4)	201 (74.2)	50 (18.5)		
Q2	270 (25.0)	13 (4.8)	204 (75.6)	53 (19.6)		
Q3	270 (25.0)	19 (7.0)	197 (73.0)	54 (20.0)		
Q4	270 (25.0)	12 (4.4)	180 (66.7)	78 (28.9)		
Total *N* (%)	1081 (100.0)	64 (6.0)	782 (72.3)	235 (21.7)		

^a^ Q4 (Quartile 4) is the richest and Q1 (Quartile 1) is the poorest. Q1: 0–3492 US$; Q2: 3493–6056 US$; Q3: 6057–10345 US$; Q4: > 1.345 US$. Note: The exchange rate between the US$ and the Chinese Yuan used in the study is 1 Yuan = 0.1445 US$.

**Table 4 ijerph-16-02543-t004:** Association between maternal education and children’s weight status (Ref. group: normal).

Variables	Model 1 (No Covariates)				Model 2 (Covariates)			
	Underweight OR (95%CI)	*p*-Value	Overweight/Obesity OR (95%CI)	*p*-Value	Underweight OR (95%CI)	*p*-Value	Overweight/Obesity OR (95%CI)	*p*-Value
Maternal education								
Junior high school and below	1.0		1.0		1.0		1.0	
High school	0.536 (0.226–1.271)	0.157	2.065 (1.370–3.113)	<0.01	0.566 (0.232–1.379)	0.210	2.170 (1.398–3.370)	<0.01
Technical college	0.229 (0.055–0.954)	0.043	2.357 (1.506–3.690)	<0.01	0.223 (0.052–0.956)	0.043	2.397 (1.478–3.887)	<0.01
College and above	0.204 (0.049–0.849)	0.029	2.251 (1.463–3.464)	<0.01	0.182 (0.041–0.812)	0.026	2.146 (1.293–3.560)	<0.01
Gender								
Boys					1.0		1.0	
Girls					0.742 (0.440–1.253)	0.264	0.542 (0.398–0.737)	<0.01
Residence								
Urban					1.0		1.0	
Rural					0.954 (0.543–1.675)	0.870	1.054 (0.756–1.469)	0.756
Children age (years)								
7–10					1.0		1.0	
11–14					1.135 (0.627–2.054)	0.676	0.976 (0.687–1.385)	0.890
15–18					0.747 (0.338–1.650)	0.471	0.446 (0.272–0.730)	<0.01
Maternal age(years)								
27–36					1.0		1.0	
37–46					1.035 (0.569–1.883)	0.910	0.778 (0.546–1.110)	0.167
47–57					0.286 (0.036–2.255)	0.235	0.964 (0.455–2.039)	0.923
Maternal employment								
No					1.0		1.0	
Yes					1.099 (0.586–2.062)	0.768	0.559 (0.388–0.805)	<0.01
Household income ^a^								
Q1					1.0		1.0	
Q2					0.653 (0.316–1.350)	0.250	1.059 (0.677–1.654)	0.802
Q3					1.050 (0.539–2.046)	0.886	1.092 (0.695–1.715)	0.702
Q4					0.907 (0.414–1.986)	0.807	1.507 (0.949–2.396)	0.082

^a^ Q4 (Quartile 4) is the richest and Q1 (Quartile 1) is the poorest. Q1: 0–3492 US$; Q2: 3493–6056 US$; Q3: 6057–10345 US$; Q4: > 1.345 US$. Note: The exchange rate between the US$ and the Chinese Yuan used in the study is 1 Yuan = 0.1445 US$.

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
