# Peer review of "Association between Maternal Education and School-Age Children Weight Status: A Study from the China Health Nutrition Survey, 2011"

_ijerph, 2019, doi:10.3390/ijerph16142543_

Round 1
Reviewer 1 Report
Explain on the tables the meaning of RMB; and in table 4 what does the values mean.
The authors should discuss the somewhat contradictory meaning of the protective effect of overweight/obesity with the fact that mothers have a paid employment: OR = 0.559 (CI: 0.338-0.805, p = 0.002) Table 4.
Author Response
Thank you for giving valuable comments for our manuscript!
Correspondingly responses were given in the reply file.and we have made revisions in the manuscript. Please see the attachment. It's my pleasure to receive your comments on any other questions regarding the manuscript.

Reviewer 2 Report
I have two major concerns about this paper. First is the strong causal language used throughout the paper that is used to describe what is actually an association. This is cross-sectional data and the “effect” of mother’s education on the weight status of school-aged children cannot be determined using this study design. This is a basic epidemiological concept and lack of understanding of this is worrisome (as is using “likelihood” to describe results from an odds ratio). Second is the lack of context for these findings – why is examining this association useful; what are the public health implications; what sorts of interventions could be designed/implemented to address concerns around maternal education and child weight status? How are these findings novel and important? A much stronger case needs to be made as to what this paper is contributing to the literature.
Overall:
Causal language needs to be removed from throughout the paper. You cannot determine the effect of mother’s education on child weight status using a cross-sectional study design. However, you can look at the association between maternal education and child weight status.
Language around reporting odds ratios needs to be corrected. If you are using an odds ratio, you need to report odds (not likelihood). E.g. from abstract, “Children with maternal education at junior high school were 2.170 times more likely to be” should be corrected to “Children with maternal education at junior high school had a 2.170 greater odds.” Unless it’s the convention of this journal, I would report effect sizes to the tenth decimal place (i.e. 2.2, not 2.170).
I appreciate that this article was likely not first written in English. However, I would recommend the article be edited by a native-English speaker to ensure language flows well and sounds less translated.
Introduction:
Lines 37-39. Please clarify what lower physical health means. Please elaborate on the complications of overweight and obese.
Clarify why it is important to determine the association between maternal weight status and school-age children’s weight status. Why is this novel and important? What sort of interventions could this lead to? How could this improve population health?
Methods:
More details on the variables collected should be included? Are they time-varying or just from a single survey? How were the co-variates coded (i.e. binary, categorical, continuous)? Were data self-reported? Via surveys? Interviews?
Tables – data categories should be clarified (i.e. age “15- “ means 15-18? Same for maternal age, etc).
Discussion:
Why is 2012 data referenced for underweight comparisons but 2010 data is referenced for overweight/obesity comparisons? Please be consistent.
Please elaborate, with clarity, on why there are such differences in Chinese vs. Western studies. You elude to this in the paragraph starting on line 156, but please be more specific and clear as to differences in findings from studies in these two populations.
Please included additional limitations, such as cross-sectional data (inability to draw causal inference), self-reported data limitations, etc.
Please elaborate on what sorts of early interventions could be helpful. Why are these findings relevant or useful? How can they impact population health? How/why are these findings important?
Author Response
Thanks for giving precious comments for our manuscript!
Correspondingly responses were given in the reply file.and we have made revisions in the manuscript. Please see the attachment. It's my pleasure to receive your comments on any other questions regarding the manuscript.
